# Morphological and Molecular Changes during Limb Regeneration of the *Exopalaemon carinicauda*

**DOI:** 10.3390/ani14050685

**Published:** 2024-02-22

**Authors:** Chaofan Xing, Mintao Wang, Zhenxiang Chen, Yong Li, Xinlei Zhou, Lei Wang, Yao Zhong, Wenjia Li, Xin Shen, Huan Gao, Panpan Wang

**Affiliations:** 1Jiangsu Key Laboratory of Marine Biotechnology, Jiangsu Ocean University, Lianyungang 222005, China; xingcf@jou.edu.cn (C.X.); wmt13930053376@163.com (M.W.); czx1826074421@163.com (Z.C.); lyde651@163.com (Y.L.); 17851371604@163.com (X.Z.); w18985044248@126.com (L.W.); 15771432379@163.com (Y.Z.); lwj6668882023@163.com (W.L.); shenthin@163.com (X.S.); huanmr@163.com (H.G.); 2Co-Innovation Center of Jiangsu Marine Bio-Industry Technology, Jiangsu Ocean University, Lianyungang 222005, China; 3Jiangsu Institute of Marine Resources Development, Jiangsu Ocean University, Lianyungang 222005, China; 4The Jiangsu Provincial Infrastructure for Conservation and Utilization of Agricultural Germplasm, Nanjing 210014, China

**Keywords:** limb regeneration, *Exopalaemon carinicauda*, morphology, transcriptome

## Abstract

**Simple Summary:**

In crustaceans, especially economic shrimp and crabs, autotomy significantly affects the survival rate and growth performance. At present, the molecular mechanism of the limb regeneration process of economic crustaceans is rarely studied. In this study, we used the pressure method to remove the pereopods of *Exopalaemon carinicauda* and recorded the regeneration process in detail. Moreover, regenerated pereopod tissue was sliced and stained with H.E. Microscopic observations revealed significant changes in the type and number of associated cells including outer epithelial cells, granulocytes, embryonic cells, and so on. This study performed a comparative transcriptome analysis of samples from different stages of limb regeneration and identified multiple differentially expressed genes that may be associated with crustacean growth or molting. The sequence and expression characteristics of the innexin gene were analyzed. Our study explored the morphological and molecular changes during limb regeneration of *E. carinicauda*, and laid a foundation for further research on molecular regulatory mechanisms.

**Abstract:**

With the increase in breeding density of *Exopalaemon carinicauda*, appendage breakage may occur, which seriously affects survival and economic benefits. To study the limb regeneration process of *E. carinicauda*, we induced autotomy of the pereopods. After a period of time, wound swelling disappeared, the pigment gradually accumulated, and a tawny film subsequently formed in the wound. The healing period of the wound occurred 24 h after autotomy, and the blastema formation stage occurred 48 h after autotomy. After 4 days of cutting, the limb buds began to differentiate, grow, and expand rapidly, and this process lasted approximately 15 days. Microscopic observations revealed significant changes in the type and number of associated cells including outer epithelial cells, granulocytes, embryonic cells, columnar epidermal cells, elongated cells, and blastoma cells, during the process from limb fracture to regeneration. A comparative transcriptome analysis identified 1415 genes differentially expressed between the J0h (0 h post autotomy) and J18h (18 h post autotomy), and 3952 and 4366 differentially expressed genes for J0 and J14d (14 days post autotomy) and J18h and J14d, respectively. Some of these genes may be related to muscle growth or molting, as indicated by the presence of *troponin C*, *chitinase*, *actin*, *innexin*, and *cathepsin L*. As a functional gene involved in epidermal formation, the mRNA expression level of the *innexin inx2* in the pereopod of *E. carinicauda* changed significantly in the experimental groups (*p* < 0.05). The results of this study contribute to existing knowledge of regeneration mechanisms in crustaceans.

## 1. Introduction

Limb regeneration is a common defense mechanism against predation [1]. Amphibians, arthropods, and mollusks have retained the ability to regenerate severed limbs during the long evolutionary process. In most crustaceans, after a limb is amputated, the wound heals to form a germ, which undergoes molting and grows into a new limb [2,3]. Studies on the regeneration of severed limbs in crabs have shown that body growth slows after limb amputation, and the cycle of the molting or molting process is prolonged [4,5]. In decapod crustaceans, molting and the regeneration of lost appendages are tightly coupled, hormonally regulated processes [6,7]. Limb regeneration is regulated by many factors, such as temperature, photoperiod, chemicals, and exogenous hormones [6,8,9]. The inhibitory effects of sodium pentachlorophenate (Na-PCP) were more pronounced in the initial phases of limb regeneration (involving wound healing, cell division, and dedifferentiation) in the grass shrimp *Palaemonetes pugio* [10]. Ecdysone regulates the ecdysone cycle; for example, in the fiddler crab (*Uca pugilator*), appendage regeneration occurs only when ecdysone titers are low [11,12]. The molecular mechanism of limb regeneration has been an important topic in regenerative science.

The regeneration process involves a series of physiological changes. Some studies have found that multiple functional genes may be involved in different stages of the regeneration process, such as the *wnt-3a* gene [13], insulin-like growth factor-I (*IGF-I*) [14], innexin [15], and fibroblast growth factor (*Fgf*) [16]. The insulin-like signaling (ILS) pathway regulates limb regeneration in *Eriocheir sinensis* by promoting muscle growth and regeneration in response to autotomy stress [17]. Fu et al. investigated protein abundance changes during limb regeneration in the swimming crab (*Portunus trituberculatus*) and identified some potential functional proteins, such as cuticle protein, myosin protein, zinc finger protein, and fibropellin protein [18]. Transcriptome sequencing has been widely used to investigate animal physiological mechanisms at the molecular level, which is most often used for analyzing differential gene expression (DGE) [19,20]. Yue et al. conducted RNA-seq sequencing on regenerated limb bud tissues of *Litopenaeus vannamei* and identified a large number of differentially expressed genes [21]. By using RNA interference (RNAi) technology to knock down the *smox* gene, a key downstream transcription factor in the activin pathway, crayfish can be induced to regenerate intact but smaller appendages after cutting [22].

*Exopalaemon carinicauda*, also known as the ridgetail white prawn, is distributed mainly on the coast of mainland China and the west coast of the Korean Peninsula [23,24]. *E. carinicauda* has the advantages of a strong reproductive ability, a short growth cycle, and a long breeding season, so it represents an important economic breeding variety in China, the annual production of which exceeds 50,000 tons, and the output value exceeds USD 420 million [25,26,27,28]. Like most other crustaceans, *E. carinicauda* has the powerful ability to regenerate severed limbs. With increasing breeding density, competition occurs among breeding individuals for food and living space, which can cause damage and the fracture of appendages. The stocking density significantly affects the growth performance, survival rate, and cheliped injury status of *Pasifastacus leniusculus* [29,30].

To study the limb regeneration process of *E. carinicauda*, we induced autotomy of the pereopods and recorded the regeneration process in detail. We performed a comparative transcriptome analysis of samples from different stages of regeneration, focusing on crustacean growth or molting-related genes. The results of this study contribute to our understanding of regeneration mechanisms.

## 2. Method

### 2.1. Samples

Healthy *Exopalaemon carinicauda* were obtained from the full-sib family constructed at the Jiangsu Key Laboratory of Marine Biotechnology (Lianyungang, China). The average body length and body weight of the *E. carinicauda* individuals were 4.5 ± 0.5 cm and 1.4 ± 0.4 g, respectively. All the samples were acclimated for two weeks with stable salinity (27–29) and pH (8.1–8.4). The shrimp were fed twice daily with commercial pellets and the filtered seawater was renewed every two days.

### 2.2. Treatment and Sampling

After rearing, four hundred healthy individuals were randomly divided into two groups: a control group (normal group) and an experimental group (limb amputation group). The individuals were divided into ten flat-bottomed rectangular tanks (80 cm × 50 cm × 30 cm). To reduce the influence of individuals and to facilitate statistical analysis, each tank was divided into separate spaces by acrylic perforated plates and each tail shrimp occupied a separate space. All individuals were given the same care, including appropriate water conditions and normal feeding.

In the limb amputation group, the pressure method was used to remove the first and third pereopods on the left side and the second and fourth pereopods on the right side for a total of 4 pereopods. Four pereopod bases of each experimental individual (control group and experimental group) were immediately frozen in liquid nitrogen for RNA extraction. Four pereopod bases were collected at 0 h post autotomy for the control group. The experimental group included samples taken 18 h and 14 days after the limb was treated. Three biological replicates were performed for each group. To meet the total RNA requirements for transcriptome sequencing, each sample included the pereopod base of no less than 20 tails of shrimp. The characteristic changes of appendage bases were observed by stereo microscope (Phenix SMZ180, Nanchang, China) at 0 h, 6 h, 24 h, 48 h, 72 h, 4 d, 6 d, 8 d, 9 d, 11 d, and 15 d after amputation.

### 2.3. Histological Analysis

To further elucidate the morphological changes that occur during the early stage of appendage regeneration, whole basal regeneration parts of the appendages were collected at 6 time points: 0 h, 6 h, 12 h, 24 h, 48 h, and 72 h. The samples were fixed, sliced, and stained with HE for microscopic observation.

Parts of the pereopod bases were rinsed with normal saline and preserved in 4% paraformaldehyde for hematoxylin–eosin (HE) staining. The pereopod bases were placed in Bouin’s fixation solution at 4 °C for 24 h and then transferred to 70% ethanol for reserve. The tissue samples were dehydrated by gradient ethanol (70%, 85%, 95%, and 100%), cleared with xylene, and embedded in paraffin. Subsequently, the paraffin-embedded tissues were sliced into 5 μm thick sections by a microtome (Leica RM2128, Wetzlar, Germany). The sections were stained with hematoxylin and eosin (H.E). The finished sections were sealed with gum for preservation, observed under an optical microscope, and photographed (Leica DM500, Wetzlar, Germany).

### 2.4. RNA Extraction and cDNA Synthesis

In the experiment, RNAiso Plus reagent (TaKaRa, Dalian, China) was used to extract the total RNA from pereopod bases of *E. carinicauda* following the manufacturer’s instructions. The Agilent Bioanalyzer 2100 (Agilent Technologies, Santa Clara, CA, USA) was used to detect the integrity of the RNA samples, and the detection indices included the RNA integrity number (RIN), 28S/18S ratio, map baseline, and 5S peak. 

After the samples were tested for qualification, approximately 3 μg of the total RNA per sample was obtained to synthesize cDNA. The nine cDNA libraries, including J0h_J1, J0h_J2, J0h_J3,, J18h _J4, J18h _J5, J18h _J6, J14d_J7, J14d_J8, and J14d_J9, were constructed using the NEBNext^®^ Ultra™ RNA Library Prep Kit for Illumina^®^ kit (NEB, Ipswich, MA, USA).

### 2.5. Transcriptome Sequencing and Assembly

Qualified libraries were sequenced on the Illumina NovaSeq 6000 sequencing platform with 150 bp paired ends. To ensure the accuracy of the sequence assembly and subsequent analysis, the raw data were filtered using Trimmomatic software (version 0.36) to obtain high-quality clean reads [31]. Trinity software (version: trinityrnaseq_r20131110) was used to assemble clean reads from nine sequenced libraries to obtain a unigene library of *E. carinicauda* [32]. Assembled unigenes were annotated against the National Center for Biotechnology Information (NCBI)-nr, NCBI-nt, Protein family (Pfam), Swiss-Prot databases, Gene Ontology (GO), and KEGG Orthology (KO) (E-value ≤ 1 × 10^–5^). The ORF coding frame information of transcripts was extracted from most unigene sequences based on alignments from the Nr and Swiss-Prot databases. For unmatched sequences, TransDecoder (https://github.com/TransDecoder/TransDecoder/wiki, accessed on 1 July 2023) was used to predict their ORF.

Then, Bowtie2 was used to align the clean reads of each sample to the assembled unigene library, and the expression level was estimated in combination with RSEM software (version 1.2.31) [33,34]. The FPKM (fragments per kilobase of transcript sequence per million base pairs sequenced) value was used to represent the expression abundance of the corresponding unigenes [35]. The DEGseq2 software package in R software (version 3.6.1) was used to screen DEGs and obtain DEG sets before and after limb amputation in *E. carinicauda* [36]. Genes were considered to be significantly differentially expressed when the |log2(fold change)| > 2 and adjusted *p*-value (padj) < 0.05 [37]. In order to explore the functional categories of differentially expressed genes, a GO and KEGG functional enrichment analysis of differentially expressed genes were performed by Goseq (version 1.14.0) and KOBAS software (version 2.0) respectively, with the *padj* < 0.05 [37,38].

### 2.6. Experimental Verification Using RT–qPCR

To confirm the accuracy of the transcriptome sequencing results, ten interesting DEGs related to growth and development were validated by real-time PCR. Primer Premier 5.0 was used to design primers based on sequence information obtained from transcriptome sequencing [39]. Primers were initially screened by ordinary PCR, and primers with nonspecific amplification were excluded (Appendix A). To further determine the amplification efficiency and specificity of the primers, cDNA samples diluted 10 times were used as templates. The cDNA samples were derived from the reverse transcription of RNA, which was the same batch of RNA used to prepare the libraries for RNAseq. The amplification efficiency of the primers was determined via fluorescence quantitative PCR using a SYBR^®^ Premix kit, and the specificity of the primers was determined according to the melting curve. Using *EF1-α* as the reference gene, the expression of DEGs was verified by fluorescence quantitative PCR, and the relative expression levels were calculated with the 2^−△△Ct^ method. There were three biological replicates and three technical replicates for each sample.

The experimental results were expressed by the mean ± standard deviation. SPSS 26.0 statistical software (SPSS Inc., Chicago, IL, USA) was used for one-way analysis of variance (ANOVA), and a multiple comparisons of Duncans was used to test the significance of the difference between the mean values, and *p* < 0.05 indicated a significant difference.

### 2.7. Homologous Cloning and Sequence Analysis

The total RNA from pereopod bases of *E. carinicauda* was extracted using RNAiso Plus reagent (TaKaRa, Dalian, China). The quality of RNA was tested using the methods (see Section 2.4). First-strand cDNA was synthesized using the PrimeScript™ II cDNA Synthesis Kit (TaKaRa, Dalian, China) following the manufacturer’s instructions. Primer sequences of the *innexin inx2* (forward 5′-ATGTATGACGTTTTCGGAAG-3′, reverse 5′-TTACAAGGGACCTTTGCCCT-3′) were designed based on our transcripts using the Primer 5 software. PCR amplification products were ligated into the pMD19T simple vector (TaKaRa, Dalian, China) and sequenced in both directions by the Shanghai Sangon Biotech Co., Ltd. (Shanghai, China). The ORF finder (https://www.ncbi.nlm.nih.gov/orffinder/, accessed on 10 July 2023) locates all open reading frames of cloned genes. The secondary structure of the amino acid sequences was predicted by the SOPMA (https://npsa-pbil.ibcp.fr/cgi-bin/npsa_automat.pl?page=npsa_sopma.html, accessed on 5 July 2023) and SMART (http://smart.embl-heidelberg.de/, accessed on 5 July 2023) software [40,41], and the tertiary structure was predicted by the SWISS-MODEL (https://swissmodel.expasy.org/, accessed on 10 July 2023) [42]. DNAMAN version 9 tool (Lynnon Biosoft, San Ramon, CA, USA) was used for multiple sequence alignment and the MEGA 7 software was used to perform a phylogenetic analysis [43].

## 3. Results

### 3.1. Morphological Observation of Limb Regeneration in E. carinicauda

The wound on the severed limb was immediately covered by a transparent film, which showed obvious swelling (Figure 1A). After a period of time, the swelling of the wound disappeared, the pigmentation gradually accumulated in the wound, and a tawny film (scab) subsequently formed in the wound, and the wound size was significantly reduced (Figure 1B). Twenty-four hours after autotomy, the scab layer at the wound site degenerated, the pigment concentration decreased, and the pigment gathered at the outer edge of the wound; moreover, the wound surface became relatively smooth, indicating complete healing of the wound (Figure 1C). This period was the healing period of the wound. Forty-eight hours after autotomy, the pigment disappeared in the middle area of the wound surface, and all of the pigment accumulated at the outer edge (Figure 1D). At this time, transparent hemiglobular protrusions protruded from the middle of the wound surface; this transparent hemiglobular projection is called a blastema, and this period was the blastema formation stage (Figure 1E). Once the blastome was generated, it could continue to grow to the distal end through cell proliferation. 

Four days post autotomy, it had grown into transparent rod-like limb buds (Figure 1F). At this time, it can be seen that the limb buds had begun to differentiate and that different segments had appeared. After 6 days of self-incision, the limb bud continued to grow and extend, the length increased significantly, and the base of the limb bud became thicker, which squeezed the wound surface and made it attach (Figure 1G). Different segments of the new limb bud differentiated significantly, and the segment shape of the new limb bud at the second step was different from that at the third, fourth, and fifth steps, while the shape of the last three pairs was similar. In addition, the color of the new limb bud became darker and was no longer transparent, indicating the growth period of the limb bud. From the 8th day to the 9th day after autotomy, the new limb grew rapidly and expanded, its size increased by three or four times compared with that of the limb bud, and all segments (ischial segment, long segment, tibia segment, tarsus segment, and toe segment) completed differentiation, especially at the distal end of the second step (Figure 1H,I). In addition, epidermal pigment cells were evident in the new limb, especially at the junction of segments, which was the growth period of the new limb. When the new limb growth reached its maximum size, the regenerative appendages stopped growing. In general, the application of the regeneration process of *E. carinicauda* was completed in approximately 15 days (Figure 1K).

### 3.2. Histological Analysis of Early-Stage Regeneration of Appendages

At 0 h after self-incision, the cells near the wound surface of the appendage exhibited a primitive state, and there were two cell distribution states: one was the outer epithelial cells, composed of a single layer of columnar cells with small nuclei and no obvious nucleoli, and the other was the cells with a loose internal arrangement (Figure 2A). From 6 h to 12 h after autotomy, a large number of cells migrated to the wound, including granulocytes and embryonic cells, among which some granulocytes came from the remote edge of the wound (Figure 2B,C). The dedifferentiation of columnar epidermal cells into granulocytes was also observed 24 h after autotomy, and elongated cells with granular cytoplasm were also observed in the epidermal cells (Figure 2D). Twenty-four hours, 48 h, and 72 h after limb amputation were the main periods of blastoma cell formation and proliferation, and the cell morphology also exhibited obvious changes (Figure 2D–F). After 24 h, blastema cells began to form; after 48 h, a large number of fibroblast-like spindle cells, namely, blastema cells, appeared on the wound surface; after 72 h, the blastema cells continued to proliferate, and blastema cells could also be observed on the distal end of the wound surface.

### 3.3. Transcriptome Sequencing and Assembly

The library of the J0 group yielded a total of 20.8 Gb of data, including 69.6 million raw data and 67.8 million clean data (Appendix A). The library of the J18h group yielded a total of 20.5 Gb of data, including 68.4 million raw data and 65.8 million clean data. There was a total of 18.1 Gb of data, including 60.1 million raw data and 57.9 million clean data for the J14d group. A total of 68,435 unigenes were obtained, with an average length of 1094 bp and an N50 of 1740 bp.

### 3.4. Functional Annotation Analysis

For functional annotation, 68,435 unigenes were searched against seven public databases, and 31,610 (46.18%) unigenes were annotated in at least one database. Using Blast2GO, 21,627 unigenes were classified into three GO categories consisting of 43 groups (Appendix A). In the biological process category, which included 26 subtypes, most genes were annotated in the cellular process (12,646), metabolic process (10,247), and biological regulation (4495) categories. The cellular component category included 5 subtypes, and most of the genes were classified into the cellular anatomical entity (10,137), intracellular (5667), and protein-containing complex (4034) categories. The molecular function category included 12 subtypes, and most unigenes were annotated as the binding (10,610), catalytic activity (8180), or transporter activity (1908).

According to the KEGG classification, a total of 10,409 unigenes were assigned to 34 pathways, which could be divided into five branches (Appendix A). Among these pathways, “signal transduction” (1291), “transport and catabolism” (1031), “translation” (1023), “endocrine system” (756), and “cell growth and death” (622) were the five pathways associated with the most genes. Several growth-related pathways, such as those related to development and regeneration, cell growth and death, carbohydrate metabolism, and amino acid metabolism, were identified.

### 3.5. Identification of DEGs

In this study, we selected the criteria *padj* < 0.05 and |log2(fold change)| > 2 to define significantly differentially expressed genes. A total of 1415 DEGs were identified between the J0h and J18h, including 180 upregulated genes and 1235 downregulated genes. Some growth-related genes, such as *xanthine dehydrogenase/oxidase-like*, *chitinase*, *trypsin*, *glucose-6-phosphatase*, and *sodium/glucose cotransporter 5-like*, were significantly differentially expressed. A total of 3952 DEGs were identified between the J0h and J14d, including 1377 upregulated genes and 2575 downregulated genes. Among these differentially expressed genes, some may be related to muscle growth or molting, such as *troponin C*, *insulin-like growth factor-binding protein*, *cuticular protein*, *actin*, and *skeletal muscle actin 6* (Table 1). A total of 4366 DEGs were identified between the J18h and J14d, including 2577 upregulated genes and 1789 downregulated genes. Several growth-related genes, such as *cathepsin B*, *cuticle proprotein*, *hsp70 binding protein*, *spatzle*, and *glucose-6-phosphatase*, were identified.

### 3.6. Functional Annotation and Pathway Assignment of DEGs

We performed GO and KEGG analyses to determine which biological processes and pathways are involved in the growth regulation of *E. carinicauda*. All DEGs between J0h and J18h were assigned to one of the 5 functional groups in the GO annotation system (Figure 3A), which included ribosome biogenesis, ribosome, extracellular region, structural molecule activity, and molecular function regulator. There were 11 functional groups between J0h and J14d, such as the carbohydrate derivative metabolic process, carbohydrate metabolic process, lipid metabolic process, and oxidoreductase activity. There were 10 functional groups between J18h and J14d, related to transmembrane transport, precursor metabolite generation, energy generation, hydrolase activity, and antioxidant activity.

The obtained DEGs were annotated to the KEGG database to identify the associated biological pathways (Figure 3B). The DEGs were mapped to 228 KEGG pathways for the J0h and J18h group and 310 pathways for the J0h and J14d and J18h and J14d groups. The DEGs in these three groups were associated with multiple KEGG pathways, such as ribosome, phagosome, lysosome, apoptosis, regulation of actin cytoskeleton, and tight junction pathways. In the J0h and J18h groups, the annotated pathways included the PI3K-Akt signaling pathway, the rap1 signaling pathway, and antigen processing and presentation. In the J0h and J14d and J18h and J14d groups, the annotated pathways included protein processing in the endoplasmic reticulum, amino sugar and nucleotide sugar metabolism, glycolysis/gluconeogenesis, and the MAPK signaling pathway.

### 3.7. Quantitative PCR to Verify DEGs

To verify the expression patterns of the unigenes in the transcriptome data, ten interesting DEGs related to growth and development were screened from the pereopod of *E. carinicauda* for qPCR analysis. These ten DEGs included the *CUGBP Elav-like family member 4*, *serine protease inhibitor*, *fatty acid binding protein*, *ferritin*, *myostatin*, *hemocyanin*, *innexin inx2*, *SIFamide*, *glucose-6-phosphate exchanger SLC37A2-like*, and *cathepsin L*. Overall, the expression trend of the same gene was consistent between the two groups determined by qPCR and RNAseq (Figure 4).

### 3.8. Structure and Phylogenetic Analysis

The open reading frame of the *innexin inx2* gene was 1089 bp (GenBank accession number: PP329397), encoding a total of 362 amino acids (Figure 5A). The molecular weight of the innexin inx2 protein of *E. carinicauda* was inferred to be 42.03 KDa and the theoretical isoelectric point was 7.35 by the ExPASy. The secondary structure of the innexin inx2 protein includes 179 α-helices, 55 extended strands, 13 beta turns, and 115 random coils. The results using the SMART online tool showed that the innexin protein had an innexin domain (Figure 5B).

The mRNA expression level of the *innexin inx2* in the pereopod of *E. carinicauda* had changed significantly in the experimental groups (*p* < 0.05) (Figure 5D). Compared with that in the control group, the mRNA expression level of the genes in the 18 h and 14 d groups was significantly greater. Compared with that in the 18 h group, the transcript level of the *innexin inx2* was significantly lower in the 14 d group. The protein sequences of the innexins from closely related organisms all showed a certain degree of similarity (Appendix A). The amino acid sequence similarity of *E. carinicauda* with *Penaeus chinensis and Penaeus japonicus* was 91.16%, and the similarity with *P. vannamei* was 90.88%. The phylogenetic tree results showed that *E. carinicauda* (Palaemonidae) and Penaeidae form a branch and are then grouped with Astacidae, Porcellanidae, Oregoniidae, Portunidae, and Varunidae. 

## 4. Discussion

The results of this experiment showed that *E. carinicauda* could regenerate and complete new limb growth by molting after limb amputation, and the time for regeneration was approximately 15 days. Current research on regeneration in arthropods has shown that the time required for regeneration varies from species to species. For example, the limb regeneration in *Macrobrachium rosenbergii* takes 10 days [44]; it takes approximately 4 weeks for the first instar Chinese eriocheir crab to regenerate its appendages [45]. Furthermore, at 30–45 days post autotomy, the new limb of *E. sinensis* fully regenerated and appeared after molting [15,17].

In this study, we observed in detail the complete regeneration process of the appendages of *E. carinicauda*. After the autotomy of *E. carinicauda*, the wound was covered by a transparent film, and then melanin gradually accumulated in the wound and formed a black film (scab) at the wound. After wound healing, 48 h post autotomy, transparent hemispherical protrusions protruded from the middle of the wound, indicating the formation of blastema. Regenerative animals can develop regenerative blastema after injury, while nonregenerative animals cannot [46,47]. After the formation of the regenerated bud base, the shrimp continued to grow to the distal end through cell proliferation, and the limb buds began to differentiate into different segments 4 days after autotomy. In *E. sinensis*, the limb bud became apparent after less than 7 days and was visible after 13 days after autotomy [15]. The new limb continued to grow, extend, and expand with water, and exhibited obvious epidermal pigmentation at the joint. However, after the first molt, a new, smaller but relatively complete limb regrew at the site of the severed limb, which is true for most crustaceans. At the microscopic level, we observed the changes in different cell types during the limb regeneration process, including granulocytes, embryonic cells, columnar epidermal cells, elongated cells, epidermal cells, and blastoma cells. Wang et al. found large numbers of granulocytes aggregated at the wound site shortly after autotomy, which is consistent with our research [15]. The function of various cell types in the process of limb regeneration remains to be further explored.

In this study, a comparative transcriptome analysis was performed, and several potential functional genes were identified. We compared the gene expression levels of the three groups of samples respectively, and found that there were significant differences in the differentially expressed genes between the groups. A total of 3952 and 4366 DEGs were identified for J18h and J14d and J0h and J14d, respectively. However, only 1415 DEGs were identified for J0h and J18h. At the same time, we found that the differentially expressed genes in the body were mainly related to energy metabolism within 18 h after limb autotomy, such as *xanthine dehydrogenase/oxidase*, *glucose-6-phosphatase*, and *sodium/glucose cotransporter 5*, and so on. In the study of *L. vannamei*, most DEGs at 12 h post autotomy are related to energy metabolism [21]. In crustaceans, regeneration of the autotomized chelipeds imposes an additional energy demand called “regeneration load”, altering energy allocation [48]. With the extension of limb amputation time, the body recruits more and more functional genes related to growth or molting, such as *cathepsin B, troponin C*, and *cuticle protein.* Cuticle proteins of *P. trituberculatus* also showed upregulated expression during limb regeneration. Wang et al. identified an expanded cuticle-related family with Chitin_bind_4 domains in the *E. sinensis* and a large number of genes from the family were differentially expressed during the limb regeneration process [15]. It needs further experimental evidence to confirm the function of potential functional genes in the limb regeneration process.

Cell–cell junctions are the basic requirements of multicellular organisms [49,50,51]. A structure connects cells through the cytoplasmic membrane, connects similar cells to tissues, and maintains relative stability with neighboring tissue cells, to maintain the coordination and integrity of the cell structure [49,52]. The cellular junctions include closed junctions, anchored junctions, and communication junctions, among which the communication junctions include gap junctions, chemical synapses, and cellular desmosomes [53,54]. Current studies have shown that the gap connections include *connexin* (Cx), *pannexin* (Panx), and *innexin* (Inx), of which innexin is found only in invertebrates [55]. Innexin genes were first discovered in Caenorhabditis and Drosophila, and subsequently reported in Annelida, Proteobacteria, Coelenterata, etc. [56]. The innexin protein is an important component involved in tissue polarity, morphology, directional movement, and electrical coupling and is involved in germ cell maintenance and differentiation. 

Studies have shown that Drosophila innexin1 is expressed mainly in the epidermis, intestine, duct system, salivary glands, cardiomyoblasts, and nervous system during embryonic development; in the larval and pupal stages, it is expressed mainly in the nervous system, adult discus, and upper belt [57,58]. During Drosophila embryonic development, *inx2* affects epithelial morphogenesis, foregut development, and the response to wingless signals [59,60]. The expression of the *innexin inx2* gene related to epidermal formation was significantly upregulated at 12 h post autotomy [21]. The results of this study showed that the expression of the *innexin inx2* gene significantly increased at 18 h and 14 days. The phylogenetic relationships of species based on innexin amino acid sequences are consistent with traditional taxonomies. Wang et al. found that six *Innexin* genes in *E. sinensis*, eight *Innexin* genes in *L. vannamei*, six *Innexin* genes in *M. rosenbergii*, and four *Innexin* genes in *Macrobrachium nipponensis* were significantly upregulated after autotomy at 1 day post autotomy [15]. The copy number of the innexin gene in *E. carinicauda* remains to be studied. Research has shown that the *Inx2* RNAi treatment induces most *E. sinensis* individuals to fail to regenerate the papilla. The function of the innexin gene in the process of limb regeneration of *E. carinicauda* remains to be further explored.

## 5. Conclusions

In this study, we explored the morphological and molecular changes during limb regeneration of the *Exopalaemon carinicauda*. We documented in detail the progress of amputation from the blastema to the limb buds. Microscopic observations revealed significant changes in the type and number of associated cells during the process of limb fracture to regeneration. By comparative transcriptome analysis, we identified several DEGs associated with muscle growth or molting and cloned the *innexin inx2* gene of *E. carinicauda*. The dynamic expression patterns and functions of these genes in the limb regeneration process need to be further studied. These results contribute to elucidating the molecular mechanisms of limb regeneration in crustaceans.

## Figures and Tables

**Figure 1 animals-14-00685-f001:**
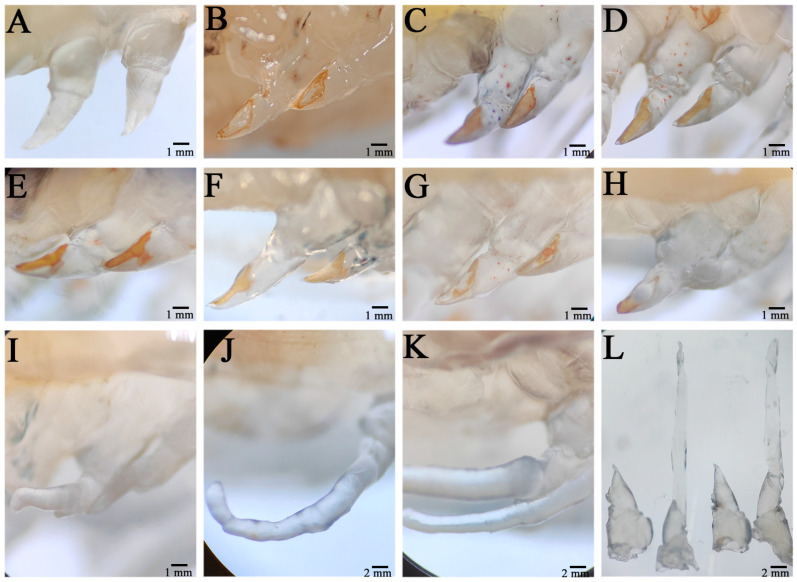
Morphological changes during limb regeneration of *E. carinicauda*. The autotomized area was photographed immediately after autotomy (**A**), 6 h after autotomy (**B**), 24 h after autotomy (**C**), 48 h after autotomy (**D**), 72 h after autotomy (**E**), 4 days after autotomy (**F**), 6 days after autotomy (**G**), 8 days after autotomy (**H**), 9 days after autotomy (**I**), 11 days after autotomy (**J**), and 15 days after autotomy (**K**). An image of 0 h versus 15 days after autotomy is also shown (**L**).

**Figure 2 animals-14-00685-f002:**
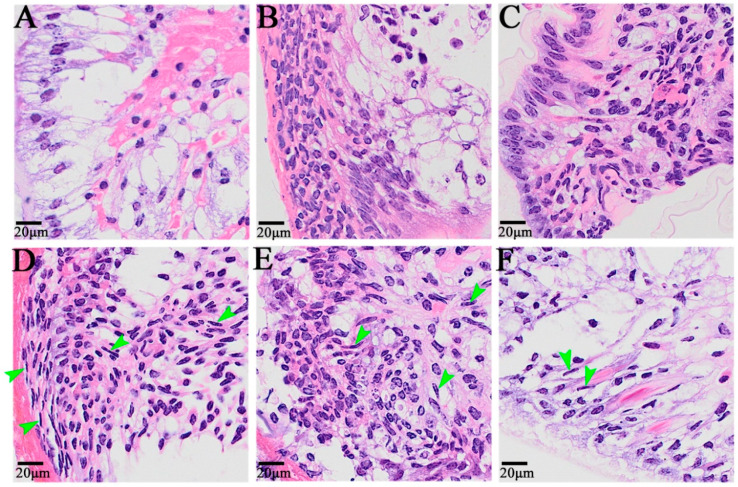
The histological symptoms during limb regeneration of *E. carinicauda*. Image (**A**): 0 h after autotomy, (**B**): 6 h after autotomy, (**C**): 12 h after autotomy, (**D**): 24 h after autotomy, (**E**): 48 h after autotomy, and (**F**): 72 h after autotomy. The nuclei were blue, and the cytoplasm was red. The green arrow refers to the blastema cells.

**Figure 3 animals-14-00685-f003:**
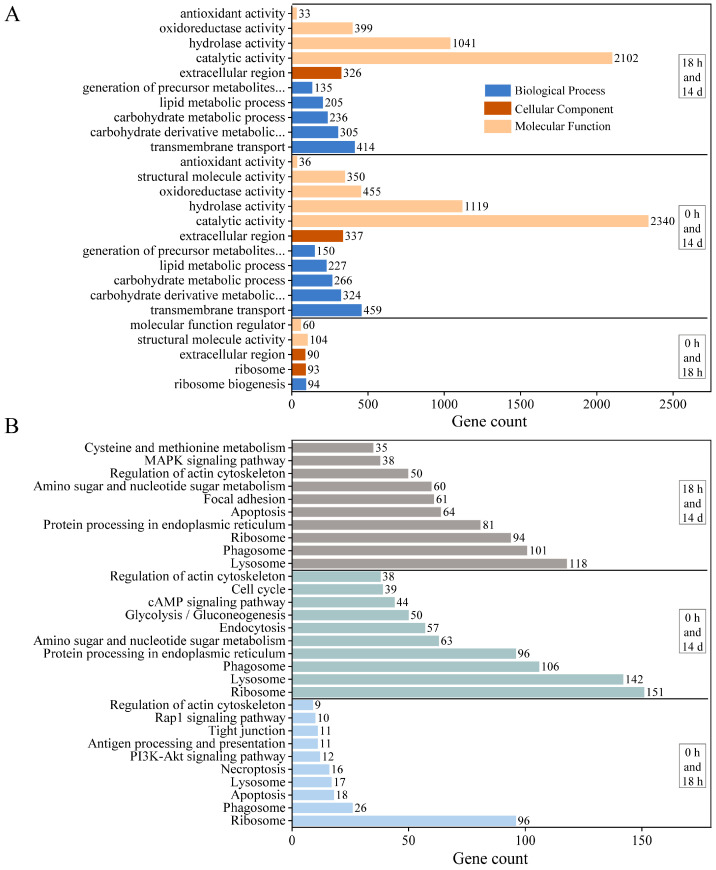
Functional annotation and pathway assignment of DEGs. (**A**) GO annotation and (**B**) KEGG pathways.

**Figure 4 animals-14-00685-f004:**
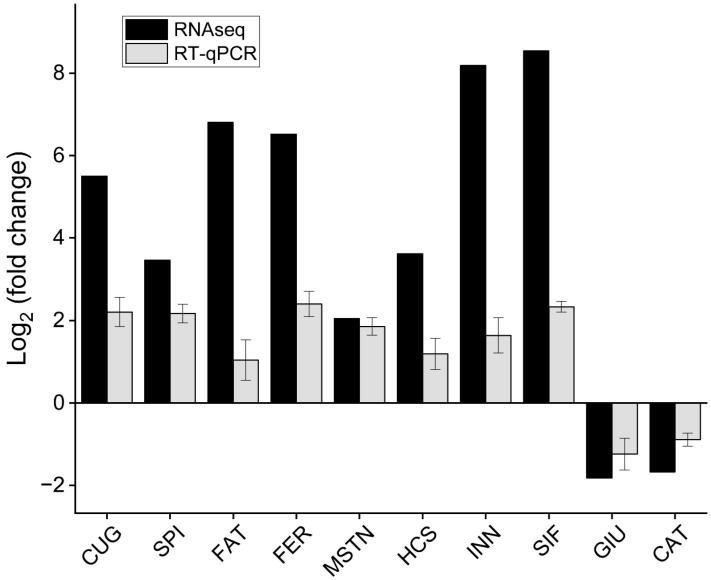
The fold change of DEGs was determined by RNA-Seq and qRT-PCR. CUG: CUGBP Elav-like family member 4, SPI: serine protease inhibitor, FAT: fatty acid binding protein, FER: ferritin, MSTN: myostatin, HCS: hemocyanin, INN: innexin inx2, SIF: SIFamide, GIU: glucose-6-phosphate exchanger SLC37A2-like, CAT: cathepsin L. CUG, MSTN, CAT, and INN were differentially expressed between J0h and J14d; SPI, FAT, FER, HCS, and SIF were differentially expressed between J18h and J14d; GIU was differentially expressed between J0h and J18h.

**Figure 5 animals-14-00685-f005:**
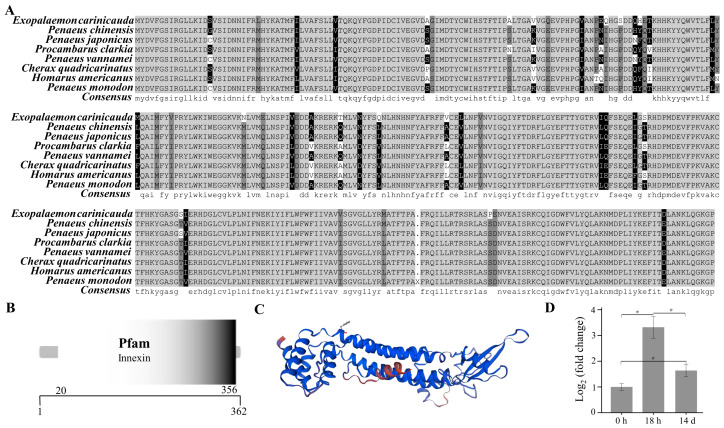
Structure and phylogenetic analysis of the innexin gene. Multiple alignments of the deduced AA sequences of the innexin proteins (**A**), structural domains (**B**), three-dimensional prediction (**C**), and the mRNA expression level of the *innexin inx2* (**D**). Each bar represents the mean ± S.D (n = 3). The asterisk “*” indicated a significant difference between groups at *p* < 0.05.

**Table 1 animals-14-00685-t001:** Candidate genes related to the growth or molting traits of *E. carinicauda*.

Groups	Gene ID	log2(FC)	pval	NR Description
J0h and J18h	Cluster-41460	5.7417	0.000	xanthine dehydrogenase/oxidase-like [*Penaeus vannamei*]
	Cluster-42390	4.7283	0.000	chitinase 1A [*Macrobrachium nipponense*]
	Cluster-42634	−2.4798	0.000	trypsin [*Penaeus vannamei*]
	Cluster-26837	−3.2764	0.000	sodium/glucose cotransporter 5-like [*Penaeus vannamei*]
	Cluster-28253	−4.4839	0.000	glucose-6-phosphate-like [*Penaeus vannamei*]
J18h and J14d	Cluster-16132	7.2655	0.000	hsp70 binding protein [*Penaeus vannamei*]
	Cluster-31691	2.2058	0.000	cuticle protein CP1499-like [*Penaeus vannamei*]
	Cluster-23596	8.7482	0.000	glucose-6-phosphatase [*Macrobrachium nipponense*]
	Cluster-34901	9.5614	0.000	cathepsin B [*Macrobrachium rosenbergii*]
	Cluster-33296	−4.5354	0.000	cuticle proprotein, partial [*Palaemon varians*]
	Cluster-12119	−5.1183	0.000	Spatzle [*Macrobrachium rosenbergii*]
J0h and J14d	Cluster-30023	8.0316	0.000	skeletal muscle actin 6 [*Rimicaris exoculata*]
	Cluster-21116	6.8346	0.000	troponin C [*Penaeus vannamei*]
	Cluster-31778	5.0923	0.000	heat shock protein 21 [*Macrobrachium rosenbergii*]
	Cluster-2529	4.2839	0.000	GATA zinc finger domain protein 14 [*Athalia rosae*]
	Cluster-17810	3.3612	0.000	zinc finger BED domain protein 5 [*Pomacea canaliculata*]
	Cluster-25392	3.2736	0.000	insulin growth factor-binding protein [*Penaeus vannamei*]
	Cluster-23228	3.0701	0.000	actin, muscle-like [*Penaeus vannamei*]
	Cluster-17462	2.5299	0.000	neprilysin-1-like [*Penaeus vannamei*]
	Cluster-26638	2.1967	0.000	heat shock protein 67B1-like [*Penaeus vannamei*]
	Cluster-33904	2.1217	0.002	zinc finger protein-like 1 [*Armadillidium vulgare*]
	Cluster-38114	−2.2901	0.001	ankyrin repeat zinc finger protein [*Penaeus vannamei*]
	Cluster-35626	−3.3186	0.000	keratin, type I cytoskeletal 9-like [*Penaeus vannamei*]
	Cluster-21125	−3.5639	0.000	cuticular protein 34 [*Eriocheir sinensis*]
	Cluster-18624	6.9349	0.000	cathepsin B [*Pandalus borealis*]

## Data Availability

Data are contained within the article and Appendix A.

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
