# Peer review of "Morphological and Molecular Changes during Limb Regeneration of the Exopalaemon carinicauda"

_animals, 2024, doi:10.3390/ani14050685_

Round 1

Reviewer 1 Report

Comments and Suggestions for Authors

Comments are in the attached PDF 

Comments on the Quality of English Language

The quality of English used in this manuscript was quite good, there were a few instances where word usage could be changed for clarity which I have tried to point out in my review. 

Author Response

Dear Editor and reviewers,

Thank you for your letter and the reviewers’ comments concerning our manuscript entitled “Morphological and molecular changes during limb regeneration of the Exopalaemon carinicauda” (animals-2837834). Those comments are valuable and very helpful for revising and improving our manuscript as well as important for guiding the significance of our research. We have read through the comments carefully and made corrections. Based on the instructions provided in your letter, we uploaded the file of the revised manuscript. The revised portions are indicated in red in the manuscript. The responses to the reviewer’s comments are as follows.

Point 1: One thing I was confused about was which tissue was being used as a ‘control’. It seems like the tissue taken at first amputation would be a good baseline, but the authors also mention a separate control group. Furthermore, the tissue being used for each sample was not clear to someone who is not an expert in shrimp anatomy. In the day-14 samples for example does the pereopod base comprise the whole pereopod that has re-grown? Clarifying what tissue and how many individuals were used in each of the datasets is very important when interpreting results of the differential expression analysis. I believe the authors do have replicate transcriptomes for each timepoint but are those transcriptomes all made from the same tissue type and number of individuals? If not, this comparison is much more difficult to interpret.

Response 1: We sincerely appreciate the valuable comments. Four pereopod bases of each experimental individual (control group and experimental group) were immediately frozen in liquid nitrogen for RNA extraction. Four pereopod bases were collected at 0 hour post autotomy for the control group. The experimental group included samples taken 18 h and 14 days after the limb was treated. Three biological replicates were performed for each group. Take group J0h as an example, which consists of three biological replicates, as J0h1, J0h2 and J0h3. To meet the total RNA requirements for transcriptome sequencing, the J0h1 sample included the pereopod base of no less than 20 tails of shrimp. This study constructed nine cDNA libraries (J0h <J1-3>, J18h <J4-6>, and J14d <J7-9>), approximately 3 μg of total RNA per sample was obtained. We have made changes in the 2.2 treatment and sampling (L99-109, 130-132).

Point 2: How were the timepoints used chosen? Are 18 hours and 14 days particularly

important in limb re-generation? It seems like the authors are saying that growth is completed in 15 days so is the 14-day timepoint supposed to represent a nearly fully re-grown limb? What does the differential expression data between the two later timepoints tell us and why is that comparison useful?

Response 2: We sincerely appreciate the professional comments. This study explored the morphological and molecular changes in the process of limb regeneration of E. carinicauda. We set 18 hours and 14 days after autotomy to explore the morphological and molecular changes in the early and late stages of limb regeneration. The expression of the gene was tissue - and time-specific. The authors believe that different stages of limb regeneration process can be regulated by multiple genes, so it is necessary to compare and analyze the differences in gene expression before and after the process. Our study found that the application of the regeneration process of E. carinicauda is completed in approximately 15 days. However, in the course of the experiment, we found that even if each tail is separated by acrylic perforated plates, there is still a high mortality rate. Therefore, 14 days after autotomy, the remaining samples were only enough for the experiment, so only a single digit of individuals survived to 15 days for morphological observation.

Point 3: I found the gene expression portion of this manuscript far too wide ranging. There is a lot of information here but not enough interpretation. I would encourage the authors to focus on a particular area. For example, the innexin protein seems to be important, and is given a lot of attention in the results, but the reason for that is not clear. A focus on this protein and a few others, and more information about them in the introduction and discussion would be nice.

Response 3: We sincerely appreciate the valuable comments. Some studies have found that multiple functional genes may be involved in different stages of the regeneration process. This study preliminarily described the sequence characteristics of innexin gene, which also laid a foundation for future functional studies. In the fourth paragraph of the discussion section, we focus on the research progress of innexin gene in the process of limb regeneration. We first introduce the background of the innexin gene as a gap junctions member, which is necessary for the general reader and even for some professional researchers in the fisheries discipline. The most basic and theoretical background introduction helps us to understand the function of this gene. The expression of the innexin inx2 gene related to epidermal formation was significantly upregulated at 12 hours post autotomy. Wang et al. found that six Innexin genes in E. sinensis, eight Innexin genes in L. vannamei, six Innexin genes in M. rosenbergii, and four Innexin genes in Macrobrachium nipponensis were significantly upregulated after autotomy at 1 day post autotomy. The function of the innexin genes in the process of limb regeneration still needs to be further studied, which is our future research direction.

Point 4: In general, the discussion needs to be expanded significantly. How does this data fit into the larger questions being asked? What is its significance? I found the results to be quite long, but very little interpretation was given particularly to the RNAseq and protein structure data later in the manuscript.

Response 4: Thanks for your friendly reminder. We have rewrote the discussion in the revised manuscript. We first described the cycle of limb regeneration in different species, and then observed the morphological changes in the process of limb regeneration. At the microscopic level, we observed the changes of different cell types during the limb regeneration process. Subsequently, we discussed the transcriptome sequencing results and found that the differentially expressed genes in the body were mainly related to energy metabolism within 18 hours after limb autotomy. Finally, we introduced the potential function of the innexin gene and the changes of its expression characteristics during limb regeneration. It is well known that the secondary and tertiary structures of genes are often closely related to their functions. This study preliminarily described the sequence characteristics of innexin gene, which also laid a foundation for future functional studies. In future studies, We will use a variety of techniques to investigate the potential functions of these genes, including RNAi, western Blot and immunohistochemical technique.

Abstract

Point 5: the abstract gives an adequate overview of the study’s purpose and main results though some clarification in necessary.

Response 5: We sincerely appreciate the valuable comments. With the increase in breeding density of Exopalaemon carinicauda, appendage breakage may occur, which seriously affects the survival and economic benefits. To study the limb regeneration process of E. carinicauda, we induced autotomy of the pereopods. The results of this study contribute to existing knowledge of regeneration mechanisms in crustaceans.

Point 6: Line 17: Unclear, is the whole 48 hours after autonomy considered the blastema formation stage?

Response 6: Thanks for your friendly reminder. Forty-eight hours after autotomy, the pigment disappeared in the middle area of the wound surface, and all of the pigment accumulated at the outer edge. At this time, transparent hemiglobular protrusions protrude from the middle of the wound surface; this transparent hemiglobular projection is called a blastema, and this period is the blastema formation stage. We have made changes in the revised manuscript (L17-18).

Point 7: Lines 19-21: Were changes in cell type discussed much in the manuscript? I feel that this section of the abstract implies that they are, but I felt this was barely touched on. Maybe expand this section of the manuscript if you have more information.

Response 7: We sincerely appreciate the professional comments. In the second paragraph of the discussion section, we focused on the complete regeneration process of the appendages of E. carinicauda. We pooled the results and found some patterns. Regenerative animals can develop regenerative blastema after injury. After the first molt, a new, smaller but relatively complete limb regrows at the site of the severed limb, which is true for most crustaceans. In our literature search, we found that there are very few descriptions about the changes of morphological characteristics in the process of limb regeneration of shrimp and crabs, and this study is a supplement to this part. At the microscopic level, we observed the changes of different cell types during the limb regeneration process, including granulocytes, embryonic cells, columnar epidermal cells, elongated cells, epidermal cells and blastoma cells. Wang et al found large numbers of granulocytes aggregated at the wound site shortly after autotomy, which is consistent with our research. The literature we searched rarely described the type and number of associated cells in shrimp and crabs.

Point 8: Lines 23-24: The meaning of J0 etc. have not been defined so these abbreviations should not be used. Instead use the timepoint full names (the time of limb removal, 18 hours after removal etc.).

Response 8: Thanks for your friendly reminder. We have made changes in the revised manuscript (L23-25).

Point 9: Line 27: Add a statement about why the change in innexin expression level is important. Does innexin need to be italicized?

Response 9: We sincerely appreciate the valuable comments. We have made changes in the revised manuscript (L27).

Introduction

Point 10: The introduction feels disjointed and while there are several statements as to the purpose of the study, they seem ununified and in some cases do not relate to the information discussed later in the paper. For example, the use of appendage fracture data as a index of stress was mentioned as a motivating factor but to my knowledge was not expanded on in later sections.

Response 10: We sincerely appreciate the valuable comments. We have rewrote the introduction in the revised manuscript.

Point 11:Line 32: Re-phrase. This sounds like limbs are only ever lost for one of these reasons.

Response 11: We sincerely appreciate the valuable comments. We have made changes in the revised manuscript (L34).

Point 12:Line 44: What is the purpose of mentioning sodium pentachlorophenate here? Both of the examples mentioned here (Palaemonetes pugio and Uca pugilator) are not well tied in with the rest of the paragraph and seem randomly placed.

Response 12: We sincerely appreciate the professional comments. In this paragraph, we introduced that limb regeneration is regulated by many factors, such as temperature, photoperiod, chemicals and exogenous hormones. The literature we searched rarely described the factors affecting the regeneration of amputated limbs of shrimp and crabs. The inhibitory effects of sodium pentachlorophenate (Na-PCP) were more pronounced on the initial phases of limb regeneration (involving wound healing, cell division and dedifferentiation) in the grass shrimp Palaemonetes pugio. In that study, author found crustacean limb regeneration can be used as a sensitive bioassay for studying the effects of chemical pollutants.

Point 13:Paragraph starting at line 49: Same comment as above. The examples given here are not clearly tied to a purpose. This section should be re-structured, and examples of previous research should clearly tie back to the purpose of this study or questions that inspired it.

Response 13: We sincerely appreciate the professional comments. We have rewrote the paragraph in the revised manuscript.

Point 14:Line 54: What is the iTRAQ approach and why does it matter here? Acronyms should be defined or not used. In this case I don’t think the specific method is necessary to mention.

Response 14: We sincerely appreciate the professional comments. We have made changes in the revised manuscript (L65).

Point 15:Line 66: This is a small thing, but it would help the reader to keep track of the target species (vs. other ones that are mentioned) if you are consistent in the name you use. Switching from “white prawn” to “white shrimp” should be standardized.

Response 15: Thanks for your friendly reminder. We have made changes in the revised manuscript (L45).

Methods

Point 16:I found the methods hard to follow. Particularly in terms of the number of individuals included in each library, and the number of transcriptomes produced per treatment. If there are multiple individuals included in each transcriptome, possibly of a different number between treatments, how can you compare expression levels between treatments? Are there replicate transcriptomes made from the same number of individuals within treatments and are these being used during the differential expression calculations? A table with metrics about the completeness of transcriptomes, their size, and the number of contigs identified per transcriptome would be very useful, even as a supplementary file if needed.

Response 16: We sincerely appreciate the valuable comments. In the control group. four pereopod bases of each experimental individual were immediately frozen in liquid nitrogen for RNA extraction. The experimental group included samples taken 18 h and 14 days after the limb was treated. Three biological replicates were performed for each group. To meet the total RNA requirements for transcriptome sequencing, ach sample included the pereopod base of no less than 20 tails of shrimp. For nine cDNA library constructions (J0h <J1-3>, J18h <J4-6>, and J14d <J7-9>), approximately 3 μg of total RNA per sample was obtained. We have added the table S2 about the summary statistics for sequencing data.

Point 17:Sections 2.5, 2.6, and 2.7 could be combined, or just sections 2.6 and 2.7. There are too many sections otherwise and it gives the impression that the methods are jumping around.

Response 17: We sincerely appreciate the valuable comments. We combined the section 2.6 and 2.7 in the revised manuscript.

Point 18:Section 2.1: This could be moved to the end of the paper with the other statements (competing interests etc.)

Response 18: Thanks for your friendly reminder. We have moved the section 2.1 in the revised manuscript (L459-462).

Point 19:Line 89: Remove “temporary”

Response 19: Thanks for your friendly reminder. We have removed in the revised manuscript (L92).

Point 20:Line 94: Dose “separate the tails of the shrimp” mean each shrimp is in a separate section of the tank? Re-word this.

Response 20: Thanks for your friendly reminder. In order to prevent cannibalism among individuals, each tank was divided into separate spaces by acrylic perforated plates and each tail shrimp occupied a separate space.

Point 21:Line 95: What is the pressure method? Do you have a citation which describes it? Section 2.3: I found the sampling process a bit confusing. Where any samples taken from the control group? It seems like the initial sample from the experimental group served as a control. Did all samples consist of four pereopods or were the 18 hour and 14 day samples taken from the re-growing pereopods of already amputated individuals? I’m assuming that once individuals were sampled they were removed from the population but that’s not stated anywhere.

Response 21: We sincerely appreciate the valuable comments. In most crustaceans, after a limb is amputated, the wound heals to form a germ, which undergoes molting and grows into a new limb. In the breeding process, when the breeding density is high or the competition for food, the fighting behavior between individuals often occurs, and the appendages are most likely to break. The appendage breakage was caused by hostile control, so we simulated external pressure and let the spine-tailed white shrimp exert their own force to cause the limb breakage. Limb breakage does not result in individual death. In the limb amputation group, the pressure method was used to force the first and third pereopods on the left side and the second and fourth pereopods on the right side for a total of 4 pereopods. The experimental group included samples taken 18 h and 14 days after the limb was treated. In order to ensure the rigor of the experiment, the samples of the three groups were taken from the same parts.

Point 22:Line 100: what does “no less than” 20 tails of shrimp mean? Were more than 20 samples taken in some cases but not used?

Response 22: We sincerely appreciate the valuable comments. Three biological replicates were performed for each group (J0h, J18h and J14d). Four pereopod bases of each experimental individual were immediately frozen in liquid nitrogen for RNA extraction. The appendage base of Exopalaemon carinicauda is millimeter-sized, very small and almost transparent. There is also the problem of varying size of the appendage base. To meet the total RNA requirements for transcriptome sequencing, ach sample included the pereopod base of no less than 20 tails of shrimp. For nine cDNA library constructions (J0h <J1-3>, J18h <J4-6>, and J14d <J7-9>), approximately 3 μg of total RNA per sample was obtained.

Point 23:I’m not sure what the three biological replicates are referring to. It seems like you had 20 or more replicates for each timepoint.

Response 23: We sincerely appreciate the valuable comments. Three biological replicates were performed for each group (J0h 1-3, J18h 1-3 and J14d 1-3). Take group J0h as an example, which consists of three biological replicates, as J0h1, J0h2 and J0h3. To meet the total RNA requirements for transcriptome sequencing, the J0h1 sample included the pereopod base of no less than 20 tails of shrimp. This study constructed nine cDNA libraries (J0h <J1-3>, J18h <J4-6>, and J14d <J7-9>).

Point 24:Line 105: Did this happen after removal from 4% PFA? I think it would be clearer to move the preservation information from section 2.3 to the respective section where that tissue is dealt with.

Response 24: We sincerely appreciate the valuable comments. We have made changes in the revised manuscript (L111-116).

Point 25:Line 107: Cleared with xylene? I don’t think tranparented is a word.

Response 25: We sincerely appreciate the professional comments. We have made changes in the revised manuscript (L119).

Point 26:Section 2.5: A bit more information is needed here. How much tissue was used, were standard procedures followed for the RNAiso Plus and are those steps available anywhere? I don’t think you need to mention the nanodrop, Qubit, and bioanalyzer since the information you’re getting from them overlap. Separate paragraphs should be used for quality control information and the way samples were mixed, which should also be clarified. Can you give an amount of RNA that was used from each individual in, for example, the 18 hour group, was combined for sequencing.

Response 26: We sincerely appreciate the professional comments. All the samples we collected were from pereopod bases of each experimental individual. We used these tools to evaluate the RNA quality during the experiment, and there was some duplication, which we have modified in the revised manuscript. Four pereopod bases of each experimental individual were immediately frozen in liquid nitrogen for RNA extraction. The appendage base of Exopalaemon carinicauda is millimeter-sized, very small and almost transparent. There is also the problem of varying size of the appendage base. To meet the total RNA requirements for transcriptome sequencing, ach sample included the pereopod base of no less than 20 tails of shrimp. Three biological replicates were performed for each group (J0h, J18h and J14d). After the samples were tested for qualified, equal amounts of total RNA from different individuals in each group were mixed in the same amount (concentration × volume). For nine cDNA library constructions (J0h <J1-3>, J18h <J4-6>, and J14d <J7-9>), approximately 3 μg of total RNA per sample (such as J0h-1) was obtained to synthesize first- and second-strand cDNA.

Point 27: Line 121: It's not clear what J0, etc. mean, these need to be defined somewhere. Are Jl-3 different libraries within the J0 group? If so this needs to be made clear. Why are three libraries used per group?

Response 27: We sincerely appreciate the valuable comments. Three biological replicates were performed for each group (J0h <J1-3>, J18h <J4-6>, and J14d <J7-9>). Take group J0h as an example, which consists of three biological replicates, as J0h1, J0h2 and J0h3. To improve the accuracy of transcriptome sequencing analysis, each treatment has three independent biological replicates, each of which builds a cDNA library.

Point 28: Section 2.6: Were any metrics used to assess the completeness of the transcriptomes? BUSCO scores? N50? These are necessary especially when comparing between groups. It's also not clear how many total individuals were combined to make one unigene library. This is necessary to compare between treatments.

Response 28: We sincerely appreciate the professional comments. clean reads for subsequent analysis were obtained after raw data filtering, sequencing error rate checking and GC content distribution checking. Summary statistics for sequencing data were added to the attachment 2. A total of 68435 unigenes were obtained, with an average length of 1094 bp and an N50 of 1740 bp (L254-255).

Point 29: Line 126: How were reads filtered? What program and parameters were used? Was any kind of analysis conducted to ensure that multiple isoforms of the same gene weren't included or that these are likely functional transcripts rather than assembly error? This can be a big issue for crustacean transcriptomes so assuring the reader that some care was taken would be good.

Response 29: We sincerely appreciate the professional comments. To ensure the accuracy of the sequence assembly and subsequent analysis, the raw da-ta were filtered using Trimmomatic software to obtain high-quality clean reads [1]. Our transcriptome sequencing data comes from professional biological companies. The problems such as multiple isoforms and assembly errors you mentioned are beyond our ability at present, and we will carry out in-depth exploration of such problems in future studies.

  • Bolger A M, Lohse M, Usadel B. Trimmomatic: a flexible trimmer for Illumina sequence data[J]. Bioinformatics, 2014, 30(15): 2114-2120.

Point 30:Line 143: Adjusted p-value should be written out. Otherwise, the reader may not know that's what padj means (I'm assuming).

Response 30: Thanks for your friendly reminder. We have made changes in the revised manuscript (L153).

Point 31:Section 2.9: More information is needed about the qPCR process. What was the input (RNA or cDNA?). If cDNA was used, was it created from a standardized amount of RNA? Where was the RNA obtained and was it from a single individual or the same RNA used to prepare the libraries for RNAseq? It's also not clear if you were comparing between treatments here or if this was only run for the J0 condition.

Response 31: We sincerely appreciate the professional comments. To ensure the batch effect of the sample, the cDNA are derived from the reverse transcription of RNA, which was the same batch of RNA used to prepare the libraries for RNAseq.

Point 32:Section 2.9: This section is very unclear. Why was cloning done? Was it to get complete sequences of transcripts obtained previously? If so, this should be explained. I don't entirely understand the purpose of the structural predictions and I don't think there was anything in the intro suggesting that structures are important to this study. This is something that should be clarified both here and in the introduction.

Response 32: Thanks for your friendly reminder. Limb regeneration is regulated by many factors, such as temperature, photoperiod and exogenous hormones. To study the limb regeneration process of E. carinicauda, we induced autotomy of the pereopods and recorded the regeneration process in detail. The regeneration process involves a series of physiological changes. Some studies have found that multiple functional genes may be involved in different stages of the regeneration process. We performed comparative transcriptome analysis of samples from different stages of regeneration, focusing on crustacean growth or molting related genes. The function of these genes in the process of limb regeneration still needs to be further studied, which is our future research direction. It is well known that the secondary and tertiary structures of genes are often closely related to their functions. This study preliminarily described the sequence characteristics of innexin gene, which also laid a foundation for future functional studies.

Point 33:Section 2.10: Which data is this referring to? I assume it's the RNAseq differential expression, but it could also be the qPCR if multiple treatments were included there. This section is very vague and needs to be substantially increased and potentially combined with each earlier section that generated data rather than included as a separate section.

Response 33: We sincerely appreciate the professional comments. This part is aimed at the analysis of qPCR results. We have moved this section to the section 2.6 (L172-175).

Results

Point 34: Lines 176-178: This should be in methods, not here.

Response 34: We sincerely appreciate the valuable comments. We have made changes in the revised manuscript.

Point 35: Lines 179 - 209: When were these observations made? Include in the methods.

Response 35: We sincerely appreciate the valuable comments. The characteristic changes of appendage base were observed by stereo microscope (Phenix SMZ180) at 0h, 6h,24h, 48h, 72h, 4d, 6d, 8d, 9d, 11d and 15 d after amputation. We have made changes in the revised manuscript (L107-109).

Point 36:Line 192: Should this be "4 days after cutting"?

Response 36: Thanks for your friendly reminder. We have made changes in the revised manuscript (L203).

Point 37:Lines 217-219: The information about collection times should be in methods along with the slide preparation and staining.

Response 37: Thanks for your friendly reminder. We have moved this information to the section 2.3 (L228).

Point 38:Section 3.3: This would read better as a table. Is the N50 an average? It seems like three datasets were generated so quality information should be provided for all of them.

Response 38: We sincerely appreciate the valuable comments. We have added the attachment 2 about summary statistics for sequencing data. Trinity (trinityrnaseq_r20131110) software was used to assemble clean reads from nine sequenced libraries to obtain a unigenes library of E. carinicauda. Due to lack of genome-wide data, the transcript sequences obtained by Trinity splicing were used as reference sequences for subsequent analysis. After Corset hierarchical clustering, the longest Cluster sequence is obtained for subsequent analysis. The length of transcript and cluster sequence were counted respectively. Thus, the N50 here is used to evaluate the transcript quality after all library splicing of E. carinicauda.

Point 39:Section 3.5: Was there any overlap in differentially regulated genes? For example, in the J0 to J18 category compared to the J0 to J14d.

Response 39: We sincerely appreciate the professional comments. It is normal to have duplicate differentially expressed genes between different comparison groups. After all, the same gene can play a role at different stages.

Point 40:Section 3.7: The DEG's chosen should be in methods. It's still not clear what is being compared here. Figure 4 has fold change but between which two time points or samples? In general validating expression patterns with qPCR is a good practice, but maybe it would have made more sense in a study more focused on specific groups of genes. In this case you have demonstrated that the RNAseq data and qPCR data align but only in a very small subset of genes. It might make sense to focus more on these genes or the functional groups they belong to rather than the very wide focus the gene expression portion of this manuscript currently has.

Response 40: We sincerely appreciate the professional comments. To confirm the accuracy of the transcriptome sequencing results, ten interesting DEGs related to growth and development were validated by real-time PCR. Therefore, the main purpose of fluorescent quantitative PCR is to verify the results of transcriptome sequencing.

Point 41:Line 320: Is this the control group that did not get limbs removed? How/when was data obtained from that group?

Response 41: We sincerely appreciate the valuable comments. Four pereopod bases of each experimental individual (control group and experimental group) were immediately frozen in liquid nitrogen for RNA extraction. The experimental group included samples taken 18 h and 14 days after the limb was treated. Four pereopod bases were collected

at 0-hour post autotomy (0hpa) for the control group.

Point 42:Lines 322-329: It isn't clear why the sequence similarity of innexin between different species is mentioned, this needs to be expanded on in the discussion. I don't think the phylogenetic tree or comparison between the gene and species tree adds to your analysis.

Response 42: We sincerely appreciate the valuable comments. In the introduction we describe the regeneration process involves a series of physiological changes and some studies have found that multiple functional genes may be involved in different stages of the regeneration process. In this study, we performed comparative transcriptome analysis of samples from different stages of regeneration, focusing on crustacean growth or molting related genes. The function of these genes in the process of limb regeneration still needs to be further studied, which is our future research direction. Phylogenetic analysis using molecular data such as DNA sequence for genes and amino acid sequence for proteins is very common not only in the field of evolutionary biology but also in the wide fields of molecular biology. This study preliminarily described the sequence characteristics of innexin gene, which also laid a foundation for future functional studies. The function and copy number of innexin gene in E. carinicauda remains to be further explored.

Discussion and Conclusions

Point 43:The discussion needs to be expanded significantly. Several of the results were not discussed further, including the histology or qPCR, and the important results from others were not adequately explained such as the functional annotation and differential gene expression. The phylogenetic and structural analyses were focused on innexin, so information about why this protein was selected from the DEGs as the focus of the study would be useful. Overall, I would like to have a clear idea about why this study was performed and what its most important findings were after reading the discussion.

Response 43: We sincerely appreciate the professional comments. We have carefully revised the discussion section by deleting the first paragraph and adding the third paragraph. In the third paragraph, we focused on the differential expression genes screened in this study, and discussed and analyzed them in combination with other studies. In the fourth paragraph, we focus on the research progress of innexin gene in the process of limb regeneration. This study preliminarily described the sequence characteristics of innexin gene, which also laid a foundation for future functional studies. Of course, this study only explored the morphological and molecular changes in the process of limb regeneration of E. carinicauda, lacking the exploration of the mechanism behind the changes, which is also the direction of our future research.

Point 44:Line 351: There needs to be a clear statement about which genes were identified as being potentially important in limb re-generation, or if those genes were already identified then which ones were differentially expressed. I think this information is present in the results, but it needs to be emphasized and discussed further.

Response 44: We sincerely appreciate the valuable comments. We have carefully revised the discussion section by deleting the first paragraph and adding the third paragraph. In the third paragraph, we focused on the differential expression genes screened in this study, and discussed and analyzed them in combination with other studies. We found that the differentially expressed genes in the body were mainly related to energy metabolism within 18 hours after limb autotomy, such as xanthine dehydrogenase/oxidase, glucose-6-phosphatase, and sodium/glucose cotransporter 5, and so on. In the study of L. vannamei, most DEGs at 12 hours post autotomy (12 hpa) are related to energy metabolism. In crustaceans, Regeneration of the autotomized chelipeds imposes an additional energy demand called “regeneration load”, altering energy allocation. In the fourth paragraph, we focus on the research progress of innexin gene in the process of limb regeneration.

Point 45:Line 353: Change "duration of life" to "time for regeneration" or similar.

Response 45: We really appreciate your suggestions. We have made changes in the revised manuscript (L360).

Point 46:Line 356: You mention the time for re-growth in other species but is it fair to compare between pereopods and legs, or other appendages? Is this timeline the same no matter which appendage is removed? I'm not sure this information is useful.

Response 46: We sincerely appreciate the valuable comments. Current research on regeneration in arthropods has shown that the time required for regeneration varies from species to species. We deleted the reference to Gryllus bimaculatus in the revised manuscript.

Point 47:Lines 361 -374: This paragraph repeats what was mentioned in the results. Is this information novel, does it tell us something important about developmental processes or survival of this species? More discussion/interpretation of the results is needed.

Response 47: We sincerely appreciate the valuable comments. In the second paragraph, we focused on the complete regeneration process of the appendages of E. carinicauda. We pooled the results and found some patterns. Regenerative animals can develop regenerative blastema after injury. After the first molt, a new, smaller but relatively complete limb regrows at the site of the severed limb, which is true for most crustaceans. In our literature search, we found that there are very few descriptions about the changes of morphological characteristics in the process of limb regeneration of shrimp and crabs, and this study is a supplement to this part. At the microscopic level, we observed the changes of different cell types during the limb regeneration process, including granulocytes, embryonic cells, columnar epidermal cells, elongated cells, epidermal cells and blastoma cells. Wang et al found large numbers of granulocytes aggregated at the wound site shortly after autotomy, which is consistent with our research. The function of various cell types in the process of limb regeneration remains to be further explored.

Point 48:Lines 378- 389: Are the types of cellular junctions important to list?

Response 48: We sincerely appreciate the professional comments. In the fourth paragraph, we focus on the research progress of innexin gene in the process of limb regeneration. We first introduce the background of the innexin gene as a gap junctions member, which is necessary for the general reader and even for some professional researchers in the fisheries discipline. The most basic and theoretical background introduction helps us to understand the function of this gene. This study preliminarily described the sequence characteristics of innexin gene, which also laid a foundation for future functional studies.

Point 49:Line 393: Here you refer to the 12-hpa group which I don't remember being mentioned before. Was this from the cited paper? Acronyms require an explanation, in this case I don't think it's worth explaining but instead don't use the acronym, just explain why the study is important and how it relates to your own results.

Response 49: We sincerely appreciate the valuable comments. The expression of the innexin inx2 gene related to epidermal formation was significantly upregulated at 12 hours post autotomy [2]. This study preliminarily described the sequence characteristics of innexin gene, which also laid a foundation for future functional studies.

  • Yue, W.; Yuan, R.; Jang, D.; Guo, X.; Li, F.; Qian, X. Transcriptome analysis of Litopenaeus vannamei during the early stage of limb regeneration process. Israeli Journal of Aquaculture - Bamidgeh 2023, 75, 1-10.

Point 50:Lines 393 -398: This section should be re-written to focus on the study presented here. It's not easy to tell which information is coming from other studies. The issue mentioned above with acronyms is also repeated with "dpa”.

Response 50: We sincerely appreciate the valuable comments. We have made changes in the revised manuscript (L421-432).

Point 51:Line 401: The method of pereopod removal does not need to be mentioned here. Instead focus on the larger picture.

Response 51: We sincerely appreciate the professional comments. We have made changes in the revised manuscript (L434-436).

Figures

Point 52:Clearer labeling of figures with a title, before delving into sub-parts, is needed throughout.

Response 52: We sincerely appreciate the valuable comments. We have added the titles of figure 1, figure 2, figure 3 and figure 5.

Point 53:Figure 1: A picture prior to amputation would be helpful for the reader to see that the limb has indeed re-grown to its previous size (if this is the case). Scale bars would also be beneficial.

Response 53: Thanks for your friendly reminder. We have added scale bars of the figure 1 in the revised manuscript.

Point 54:Figure 3: Do the colors have meaning in panel B? I think they can't have the same meaning as panel A so it would be best not to use the same colors. I would also encourage you to stay away from green and red since they are not useful for colorblind people.

Response 54: Thanks for your friendly reminder. We have recreated the figure 3 in the revised manuscript.

Point 55:Figure 4: This caption needs a lot more explanation. It's not clear where this data is from and what the fold change shown here is between.

Response 55: Thanks for your friendly reminder. Figure 4. The fold change of DEGs was determined by RNA-Seq and qRT-PCR. CUG: CUGBP Elav-like family member 4, SPI: serine protease inhibitor, FAT: fatty acid binding protein, FER: fer-ritin, MSTN: myostatin, HCS: hemocyanin, INN: innexin inx2, SIF: SIFamide, GIU: glu-cose-6-phosphate exchanger SLC37A2-like, CAT: cathepsin L. CUG, MSTN, CAT and INN were identified from J0h & J14d; SPI, FAT, FER, HCS and SIF were identified from J18h & J14d; GIU was identified from J0h & J18h.

Point 56:Figure 5: This is a nice figure and contains useful information, but I think it would be much better if the importance of this protein was established a lot earlier in the manuscript.

Response 56: We sincerely appreciate the valuable comments. Limb regeneration is a complex process, and it is impossible to confidently elucidate associated mechanisms exclusively by investigating the expression patterns of single genes. Moreover, genes associated with crab limb regeneration have received far less attention, and few investigations have focused on the mechanisms involved in regeneration at the protein level in marine organisms. This study preliminarily described the sequence characteristics of innexin gene, which also laid a foundation for future functional studies.

Point 57:Figure 6: This figure doesn't serve a clear purpose and should be removed or possibly moved to supplementary info if the authors focus more on this analysis in subsequent drafts. It's not clear what protein sequences are used.

Response 57: We sincerely appreciate the valuable comments. In this paper, we studied the innexin gene sequence, including homologous sequence alignment, secondary structure, tertiary structure, expression difference and phylogenetic relationship, which are the necessary basic introduction to a certain gene information. NJ Phylogenetic tree analysis based on protein sequences from the NCBI databases. In future studies, functional studies of this gene will be further carried out and other experimental techniques including RNAi, western Blot and immunohistochemical technique will be used. The specific function of this gene and other proteins interacting with it in the process of limb regeneration were further explored. We have moved the figure to supplementary info as the figure S3.

Reviewer 2 Report

Comments and Suggestions for Authors

The article “Morphological and molecular changes during limb regeneration of the Exopalaemon carinicauda” by Xing et al. is dealing with a very interesting topic with a vast applied potential. In this study, they used the pressure method to force the pereopods of E. carinicauda and documented in detail the progress of amputation from the blastema to the limb buds. Then, they identified several DEGs associated with muscle growth or molting and cloned the innexin inx2 gene of E. carinicauda by comparative transcriptome analysis. These results contribute to elucidating the molecular mechanisms of limb regeneration in crustaceans. However, some major issues compromised the quality of this MS. I suggest this article be received after some revisions.

Introduction

The introduction of E. carinicauda in line 62 should be moved to the second paragraph, and provide additional background, including production, origin, culture status, genes, and so on.

Add some background on transcriptome sequencing and RNAi, and cite references.

Some genes and pathways such as EcR, MIH, ILS, and smox are described in the third paragraph, do those appear in your transcriptome sequencing results? If not please don't mention them.

Methods

If the ‘Result’ involves the analysis of genes, the ‘Methods’ needs to be supplemented with relevant information such as the URL of the analysis and the tools used for graphing, and so on.

The methods can be referred to in this article ‘Molecular cloning and expression of MnGST-1 and MnGST-2 from oriental river prawn, Macrobrachium nipponense, in response to hypoxia and reoxygenation’

Results

Based on the ‘Method’ in line 121, the groups of your transcriptome are ‘J0’, ‘J18h’ and ‘J14d’. Therefore, Table 1 needs to be recreated. The new table needs to be marked which comparison groups screened for which genes like Figure3.

In addition, the graphing of Table 4 and the selection of DEGs were incorrect. Firstly, the selection of DEGs for validating transcriptome accuracy should be randomized and no tendencies. Also, the Foldchange expression trend of the selected DEGs should be similar to Figure 5D which in the three comparison groups. Finally, the figure notes should explain the abbreviations for these genes. It can be referred to in this article ‘17β-Estradiol Induced Sex Reversal and Gonadal Transcriptome Analysis in the Oriental River Prawn (Macrobrachium nipponense): Mechanisms, Pathways, and Potential Harm’

Gene names need to be italicized in line 266,304 311and so on.

Has the gene sequence of this gene been uploaded to NCBI? What's its accession number?

Discussion’

The discussion should be rewritten. The first paragraph should be deleted, it repeats the content of the third paragraph. This section should discuss your transcriptome results in detail. You need read more articles about the transcriptome. In addition to focusing on some differential genes, differential pathways should also be discussed. You can propose your conjecture based on the differences between the periods 'J0', 'J18h' and 'J14d' in conjunction with the GO and KEGG results. The fourth paragraph should be deleted; in fact, for the innexin gene you have not performed functional studies, so discussing it too much is unnecessary.

Comments on the Quality of English Language

Minor editing of English language required

Author Response

Dear Editor and reviewers,

Thank you for your letter and the reviewers’ comments concerning our manuscript entitled “Morphological and molecular changes during limb regeneration of the Exopalaemon carinicauda” (animals-2837834). Those comments are valuable and very helpful for revising and improving our manuscript as well as important for guiding the significance of our research. We have read through the comments carefully and made corrections. Based on the instructions provided in your letter, we uploaded the file of the revised manuscript. The revised portions are indicated in red in the manuscript. The responses to the reviewer’s comments are as follows.

Introduction

Point 1: The introduction of E. carinicauda in line 62 should be moved to the second paragraph, and provide additional background, including production, origin, culture status, genes, and so on.

Response 1: We sincerely appreciate the valuable comments. We have made changes in the revised manuscript (L40-49). Exopalaemon carinicauda, also known as the ridgetail white prawn, is distributed mainly on the coast of mainland China and the west coast of the Korean Peninsula [1,2]. E. carinicauda has the advantages of strong reproductive ability, a short growth cycle and a long breeding season, so it represents an important economic breeding variety in China, the annual production of which exceeds 50, 000 tons, and the output value exceeds US$ 420 million [3-6].

References:

  • Gao, H.; Ma, H.; Sun, J.; Xu, W.; Gao, W.; Lai, X.; Yan, B. Expression and function analysis of crustacyanin gene family involved in resistance to heavy metal stress and body color formation in Exopalaemon carinicauda. Journal of Experimental Zoology Part B: Molecular and Developmental Evolution 2021, 336, 352-363.
  • Hu, G.; Wang, W.; Xu, K.; Wang, C.; Liu, D.; Xu, J.; Yan, B.; Ji, N.; Gao, H. Transcriptomic and Metabolomic Analyses of Palaemon carinicauda Hepatopancreas in Response to Enterocytozoon hepatopenaei (EHP) Infection. Fishes 2023, 8, 92.
  • Xing, C.; Xiong, J.; Xie, S.; Guo, H.; Hua, S.; Yao, Y.; Zhu, J.; Yan, B.; Shen, X.; Gao, H. Comparative transcriptome and gut microbiota analysis of Exopalaemon carinicauda with different growth rates from a full-sib family. Aquaculture Reports 2023, 30, 101580.
  • Wang, C.; Han, W.; Cheng, W.; Liu, D.; Wang, W.; Yan, B.; Gao, H.; Hu, G. Impact of Ocean Acidification on the Gut Histopathology and Intestinal Microflora of Exopalaemon carinicauda. Animals 2023, 13, 3299.
  • Shen, S.; Hu, J.; Shen, Q.; Chen, H.; Gao, H.; Lai, X. Cloning of notch1 and its role in the growth and development of Exopalaemon carinicauda. Aquaculture Reports 2023, 30, 101537.
  • Xin-Zheng, L.; Rui-Yu, L.; Xiang-Qiu, L. The zoogeography of Chinese Palaemonoidea fauna. Biodiversity Science 2003, 11, 393.

Point 2: Add some background on transcriptome sequencing and RNAi, and cite references.

Response 2: We really appreciate your suggestions. Transcriptome sequencing has been widely used to investigate animal physiological mechanisms at the molecular level, , which is most often used for analyzing differential gene expression (DGE) [7,8]. RNA sequencing (RNA-seq) was developed more than a decade ago [9] and since then has become a ubiquitous tool in molecular biology that is shaping nearly every aspect of our understanding of genomic function. Yue et al. conducted RNA-seq sequencing on regenerated limb bud tissues of Litopenaeus vannamei and identified a large number of differentially expressed genes [10]. In the few years since the discovery of RNA interference (RNAi), it has become clear that this process is ancient [11]. Genetic studies have expanded the biology of RNAi to cosuppression, transposon silencing, and the first hints of relationships to regulation of translation and development. Practical applications of this knowledge have allowed rapid surveys of gene functions. This study performed comparative transcriptome analysis of samples from different stages of regeneration and identified multiple differentially expressed genes that may be associated with crustacean growth or molting. This paper does not involve RNAi related research. We added the full name- RNA interference of RNAi. More background on RNAi might be superfluous. In future studies, We will use a variety of techniques to investigate the potential functions of these genes, including RNAi, western Blot and immunohistochemical technique. We have made changes in the revised manuscript (L65-73).

References:

  • Rani, B.; Sharma, V. Transcriptome profiling: methods and applications-A review. Agricultural Reviews 2017, 38, 271-281.
  • Stark, R.; Grzelak, M.; Hadfield, J. RNA sequencing: the teenage years. Nature Reviews Genetics 2019, 20, 631-656.
  • Emrich S J, Barbazuk W B, Li L, et al. Gene discovery and annotation using LCM-454 transcriptome sequencing[J]. Genome research, 2007, 17(1): 69-73.
  • Yue, W.; Yuan, R.; Jang, D.; Guo, X.; Li, F.; Qian, X. Transcriptome analysis of Litopenaeus vannamei during the early stage of limb regeneration process. Israeli Journal of Aquaculture - Bamidgeh 2023, 75, 1-10.
  • Fire A, Xu S Q, Montgomery M K, et al. Potent and specific genetic interference by double-stranded RNA in Caenorhabditis elegans[J]. nature, 1998, 391(6669): 806-811.

Point 3: Some genes and pathways such as EcR, MIH, ILS, and smox are described in the third paragraph, do those appear in your transcriptome sequencing results? If not please don't mention them.

Response 3: We sincerely appreciate the valuable comments. This study aims to explore morphological and molecular changes during limb regeneration of the Exopalaemon carinicauda. In the introduction, it is necessary to introduce the current research progress of genes related to limb regeneration. The regeneration process involves a series of physiological changes. Some studies have found that multiple functional genes may be involved in different stages of the regeneration process, such as the wnt-3a [12], the insulin-like growth factor-I (IGF-I) [13], and the fibroblast growth factor (Fgf) [14]. The regeneration capacities are different in various organisms, based mainly on sex, age, and different development stages. There are also some differences in the genes regulating limb regeneration in different species. We performed comparative transcriptome analysis of samples from different stages of regeneration, focusing on crustacean growth or molting related genes.

References:

  • Yokoyama, H.; Ogino, H.; Stoick-Cooper, C.L.; Grainger, R.M.; Moon, R.T. Wnt/β-catenin signaling has an essential role in the initiation of limb regeneration. Developmental biology 2007, 306, 170-178.
  • Koriyama, Y.; Homma, K.; Sugitani, K.; Higuchi, Y.; Matsukawa, T.; Murayama, D.; Kato, S. Upregulation of IGF-I in the goldfish retinal ganglion cells during the early stage of optic nerve regeneration. Neurochemistry international 2007, 50, 749-756.
  • Shibata, E.; Yokota, Y.; Horita, N.; Kudo, A.; Abe, G.; Kawakami, K.; Kawakami, A. Fgf signalling controls diverse aspects of fin regeneration. Development 2016, 143, 2920-2929.

Methods

Point 4: If the ‘Result’ involves the analysis of genes, the ‘Methods’ needs to be supplemented with relevant information such as the URL of the analysis and the tools used for graphing, and so on. The methods can be referred to in this article ‘Molecular cloning and expression of MnGST-1 and MnGST-2 from oriental river prawn, Macrobrachium nipponense, in response to hypoxia and reoxygenation’

Response 4: We really appreciate your suggestions. This study performed comparative transcriptome analysis of samples from different stages of regeneration and identified multiple differentially expressed genes that may be associated with crustacean growth or molting. The sequence and expression characteristics of innexin gene were analyzed. We described homologous cloning and sequence analysis of the innexin gene in the 2.7 of the methods, including the URL and references of the tools.

Results

Point 5: Based on the ‘Method’ in line 121, the groups of your transcriptome are ‘J0’, ‘J18h’ and ‘J14d’. Therefore, Table 1 needs to be recreated. The new table needs to be marked which comparison groups screened for which genes like Figure3.

Response 5: We really appreciate your suggestions. We have recreated the table 1 in the revised manuscript.

Point 6: In addition, the graphing of Table 4 and the selection of DEGs were incorrect. Firstly, the selection of DEGs for validating transcriptome accuracy should be randomized and no tendencies. Also, the Foldchange expression trend of the selected DEGs should be similar to Figure 5D which in the three comparison groups. Finally, the figure notes should explain the abbreviations for these genes. It can be referred to in this article ‘17β-Estradiol Induced Sex Reversal and Gonadal Transcriptome Analysis in the Oriental River Prawn (Macrobrachium nipponense): Mechanisms, Pathways, and Potential Harm’

Response 6: We sincerely appreciate the valuable comments. To verify the expression patterns of the unigenes in the transcriptome data, ten interesting DEGs related to growth and development were screened from the pereopod of E. carinicauda for qPCR analysis. We studied the paper you recommended carefully. The nine genes selected by the authors showed significant differences in all comparison groups. The purpose of this study was to verify the expression patterns of differentially expressed genes. Not all of the ten differentially expressed genes we randomly selected showed significant differences among the three comparison groups. CUG, MSTN, CAT and INN were identified from J0h & J14d; SPI, FAT, FER, HCS and SIF were identified from J18h & J14d; GIU was identified from J0h & J18h. Overall, the expression trend of the same gene was consistent between the two groups determined by qPCR and RNAseq. The experimental results have reached our verification purpose. We have added the abbreviations for these genes in the revised manuscript (L322-327).

Point 7: Gene names need to be italicized in line 266,304 311and so on.

Response 7: We really appreciate your suggestions. We have made changes in the revised manuscript (L278-286, 316-318, 329, 336, 340).

Point 8: Has the gene sequence of this gene been uploaded to NCBI? What's its accession number?

Response 7: Thanks for your friendly reminder. The gene had been uploaded to the NCBI database with the accession number PP329397.

Discussion

Point 9: The discussion should be rewritten. The first paragraph should be deleted, it repeats the content of the third paragraph. This section should discuss your transcriptome results in detail. You need read more articles about the transcriptome. In addition to focusing on some differential genes, differential pathways should also be discussed. You can propose your conjecture based on the differences between the periods 'J0', 'J18h' and 'J14d' in conjunction with the GO and KEGG results. The fourth paragraph should be deleted; in fact, for the innexin gene you have not performed functional studies, so discussing it too much is unnecessary.

Response 9: We sincerely appreciate the valuable comments. We have carefully revised the discussion section by deleting the first paragraph and adding the third paragraph. In the second paragraph, we focused on the complete regeneration process of the appendages of E. carinicauda. We pooled the results and found some patterns. Regenerative animals can develop regenerative blastema after injury. After the first molt, a new, smaller but relatively complete limb regrows at the site of the severed limb, which is true for most crustaceans. In the third paragraph, we focused on the differential expression genes screened in this study, and discussed and analyzed them in combination with other studies. We found that the differentially expressed genes in the body were mainly related to energy metabolism within 18 hours after limb autotomy, such as xanthine dehydrogenase/oxidase, glucose-6-phosphatase, and sodium/glucose cotransporter 5, and so on. In the study of L. vannamei, most DEGs at 12 hours post autotomy (12 hpa) are related to energy metabolism. In crustaceans, Regeneration of the autotomized chelipeds imposes an additional energy demand called “regeneration load”, altering energy allocation. In the fourth paragraph, we focus on the research progress of innexin gene in the process of limb regeneration. In the fourth paragraph, we focus on the research progress of innexin gene in the process of limb regeneration. This study preliminarily described the sequence characteristics of innexin gene, which also laid a foundation for future functional studies. Of course, this study only explored the morphological and molecular changes in the process of limb regeneration of E. carinicauda, lacking the exploration of the mechanism behind the changes, which is also the direction of our future research.

Comments on the Quality of English Language

Point 10: Minor editing of English language required

Response 10: We sincerely appreciate the valuable comments. The manuscript has been polished by American Journal Experts (AJE).

Round 2

Reviewer 1 Report

Comments and Suggestions for Authors

Comments are attached 

Comments on the Quality of English Language

I found the paper understandable. The issues with English language apply more in terms of readability and clarity of the text. Verbs are not always correctly conjugated and there are some mix-ups between plural and singular. The word "force" was used in a way that I am not familiar with. 

Author Response

Dear Editor and reviewers,

Thank you for your letter and the reviewers’ comments concerning our manuscript entitled “Morphological and molecular changes during limb regeneration of the Exopalaemon carinicauda” (animals-2837834). Those comments are valuable and very helpful for revising and improving our manuscript as well as important for guiding the significance of our research. We have read through the comments carefully and made corrections. Based on the instructions provided in your letter, we uploaded the file of the revised manuscript. The revised portions are indicated in green in the manuscript. The responses to the reviewer’s comments are as follows.

I appreciate the changes the authors have made to their manuscript and the care they took in their responses to my and other comments. I found the manuscript to be improved though there were still some portions of the study that I found difficult to understand. I have highlighted those areas of confusion in my comments below and I believe they will be easy to rectify.

Summary

Point1:Line 16: The word “force” is not clearly used here. Does it mean remove? This should be changed or clarified throughout the manuscript.

Response 1: We sincerely appreciate the valuable comments. We have made changes in the revised manuscript (L16 and L113).

Introduction

Point2:More information was added but the introduction is still a little disorganized. Each paragraph should have a clear topic sentence and focus on a single subject. Some of the examples used in this section do not connect to the rest of the study and are distracting.

Response 2: We sincerely appreciate the professional comments. We have rewrote the introduction section in the revised manuscript. We first described the limb regeneration affects the growth and molting of crustaceans and introduced the influencing factors of limb regeneration. The molecular mechanism of limb regeneration has been an important topic in regenerative science. Subsequently, we review the current research progress on the regulatory mechanisms of limb regeneration. Finally, we introduced  Exopalaemon carinicauda, including production, origin and culture status, and explained the purpose of this study.

Point3:Lines 81-82: Re-phrase. It’s not clear what is most often used.

Response 3: We sincerely appreciate the valuable comments. We have made changes in the revised manuscript (L16 and L111).

Point4:Line 89: Is this example connected to the results or discussion? I don’t see this gene mentioned anywhere else.

Response 4: Thanks for your friendly reminder. We have removed in the revised manuscript (L78).

Methods

Point5:Thank you for including more detail in your sampling methods. I still find the section about pereopod base removal (2.2) confusing. Perhaps it would help to describe exactly what a pereopod base is? My understanding is that it’s the tissue left after limb amputation but I’m not sure what that means for the control group since no limbs were amputated.

Response 5: We sincerely appreciate the valuable comments. Many crustaceans, especially crabs and shrimps, often have the phenomenon of self-amputation of appendages. The autotomy section occurs at a double-layer membrane between the basal segment and the seat segment of the appendage. The membrane has holes that allow blood vessels and nerves to pass through, and the autotomizer muscle is emitted from the chest body wall and inserted directly into the fracture surface [1]. The autotomizer muscle is stimulated to contract violently due to the pull of an external force or its own factors, and is disentangled by the fracture surface. The contracted membrane can block the blood vessel to prevent the outflow of blood. This self-cutting phenomenon can even be caused by the reflection activity of the single node. In this study, pereopod base is shown in the figure 1A. The severed limb is not at the base of the limb, but at the coxa.

  • McVean A R. Autotomy in Carcinus maenas (Decapoda: Crustacea)[J]. Journal of Zoology, 1973, 169(3): 349-364.
  • Frisch A J, Hobbs J P A. Effects of autotomy on long-term survival and growth of painted spiny lobster (Panulirus versicolor) on the Great Barrier Reef, Australia[J]. Marine biology, 2011, 158: 1645-1652.

Point6:Line 143: I think the composition of libraries should be stated in words. The information in parenthesis is not clear to someone who has not read this manuscript before.

Response 6: Thanks for your friendly reminder. We have made changes in the revised manuscript (L146-147).

Point7:Section 2.7: I still find this section unclear. There are steps between RNA extraction/quantification and ORF finding which seem to be missing. It’s not clear how you got from RNA to cloned sequences.

Response 7: We sincerely appreciate the valuable comments. The quality of RNA were tested using the methods mentioned in 2.4. First strand cDNA was synthesized using the PrimeScript™ II cDNA Synthesis Kit (TaKaRa, Dalian, China) following the manufacturer’s instructions. Primer sequences of the innexin inx2 (forward 5’- ATGTATGACGTTTTCGGAAG-3’, reverse 5’-TTACAAGGGACCTTTGCCCT-3’) were designed based on our transcripts using the Primer 5 software. PCR amplification products were ligated into the pMD19T simple vector (TaKaRa, Dalian, China) and sequenced in both directions by the Shanghai Sangon Biotech Co., Ltd. (Shanghai, China). We have supplemented this information in the revised manuscript (L195-201).

Results

Point8:3.1: This section should be divided into multiple paragraphs.

Response 8: Thanks for your friendly reminder. To avoid redundancy, we split it into two paragraphs based on the growth stage of limbs in the revised manuscript.

Point9:Line 240: Information about the colors in figure 2 should be moved to the figure caption.

Response 9: Thanks for your friendly reminder. We have made changes in the revised manuscript (L269).

Discussion

Point10:The authors could still do more here to connect the results back to the goal of the study. Especially as the goals relate to growth in captivity and economic value of this species.

Response 10: We sincerely appreciate the valuable comments. With increasing breeding density, competition occurs among breeding individuals for food and living space, which can cause damage and fracture of appendages. The stocking density significantly affects the growth performance, survival rate, and cheliped injury status. This study explored the morphological and molecular changes in the process of limb regeneration of E. carinicauda. The discussion section is also mainly focused on these two parts. In future studies, we also need to further investigate the effects of limb amputation on survival rate, growth rate, molting rate, etc.

Point11:Paragraph starting at line 411: Divide into multiple paragraphs.

Response 11: Thanks for your friendly reminder. We split it into two paragraphs in the revised manuscript.

Figures

Point12:Figure 4. I appreciate that information about where each gene was identified is now included. When you say a gene was identified from J18h and J14d does that mean it is differentially expressed between those timepoints? Clarify in the figure caption.

Response 12: Thanks for your friendly reminder. CUG, MSTN, CAT and INN were differentially expressed between J0h and J14d; SPI, FAT, FER, HCS and SIF were differentially expressed between J18h and J14d; GIU was differentially expressed between J0h and J18h. We have made changes in the revised manuscript (L347-349).

Comments on the Quality of English Language

Point13:I found the paper understandable. The issues with English language apply more in terms of readability and clarity of the text. Verbs are not always correctly conjugated and there are some mix-ups between plural and singular. The word "force" was used in a way that I am not familiar with. 

Response 13: We sincerely appreciate the valuable comments. The manuscript has been checked by a colleague fluent in English writing, focusing on some verbs, prepositions and the normalization of single and plural. We replaced force with "remove" in the revised manuscript.

Reviewer 2 Report

Comments and Suggestions for Authors

I have no further questions.

Author Response

We would like to express our great appreciation to you for the positive and constructive comments and suggestions.